# Radiomodulating Properties of Superparamagnetic Iron Oxide Nanoparticle (SPION) Agent Ferumoxytol on Human Monocytes: Implications for MRI-Guided Liver Radiotherapy

**DOI:** 10.3390/cancers16071318

**Published:** 2024-03-28

**Authors:** Michael R. Shurin, Vladimir A. Kirichenko, Galina V. Shurin, Danny Lee, Christopher Crane, Alexander V. Kirichenko

**Affiliations:** 1Department of Pathology, University of Pittsburgh Medical Center, Pittsburgh, PA 15213, USA; shuringv@upmc.edu; 2Department of Radiation Oncology, Allegheny Health Network Cancer Institute, Pittsburgh, PA 15224, USA; vak8238@gmail.com (V.A.K.); danny.lee@ahn.org (D.L.); 3Department of Radiation Oncology, Memorial Sloan Kettering Cancer Center, New York, NY 10065, USA; cranec1@mskcc.org

**Keywords:** magnetic iron oxide nanoparticles, monocytes, macrophages, liver cancer, biomedical application

## Abstract

**Simple Summary:**

Image-guided stereotactic body radiation therapy (SBRT), utilizing biocompatible superparamagnetic iron oxide nanoparticles (SPION), like ferumoxytol, has emerged as a non-invasive, safe, and effective therapy for liver tumors. However, the radiomodulating properties of ferumoxytol on hepatic macrophages have never been directly investigated. We showed that ferumoxytol affected human monocytes increasing their resistance to radiation-induced cell death. These findings provide the basis for mechanism-based optimization of SPION-enhanced image-guided functional treatment planning platform for reducing hepatotoxicity in patients with advanced hepatic cirrhosis undergoing liver SBRT for liver cancer before liver transplant.

**Abstract:**

Superparamagnetic iron oxide nanoparticles (SPION) have attracted great attention not only for therapeutic applications but also as an alternative magnetic resonance imaging (MRI) contrast agent that helps visualize liver tumors during MRI-guided stereotactic body radiotherapy (SBRT). SPION can provide functional imaging of liver parenchyma based upon its uptake by the hepatic resident macrophages or Kupffer cells with a relative enhancement of malignant tumors that lack Kupffer cells. However, the radiomodulating properties of SPION on liver macrophages are not known. Utilizing human monocytic THP-1 undifferentiated and differentiated cells, we characterized the effect of ferumoxytol (Feraheme^®^), a carbohydrate-coated ultrasmall SPION agent at clinically relevant concentration and therapeutically relevant doses of gamma radiation on cultured cells in vitro. We showed that ferumoxytol affected both monocytes and macrophages, increased the resistance of monocytes to radiation-induced cell death and inhibition of cell activity, and supported the anti-inflammatory phenotype of human macrophages under radiation. Its effect on human cells depended on the duration of SPION uptake and was radiation dose-dependent. The results of this pilot study support a strong mechanism-based optimization of SPION-enhanced MRI-guided liver SBRT for primary and metastatic liver tumors, especially in patients with liver cirrhosis awaiting a liver transplant.

## 1. Introduction

Hepatocellular carcinoma (HCC) is the sixth most common cancer and the third leading cause of cancer-related death worldwide [1,2]. Liver cirrhosis predisposes the development of HCC, with 80–90% of HCC cases occurring in cirrhotic livers. In the USA, chronic hepatitis C, alcohol abuse, and non-alcoholic steatohepatitis are the leading causes of hepatic cirrhosis leading to HCC [3,4]. The high prevalence of hepatic cirrhosis and portal hypertension in patients with HCC markedly limits the choice of curative liver resection: only 15–30% of HCC patients with cirrhosis are eligible for curative partial hepatectomy [5,6]. Selective internal radiation therapy (SIRT) or radioembolization is one of the targeted treatments for unresectable liver tumors where radioactive microspheres are infused via the hepatic artery for internal tumor irradiation. Although it is generally considered efficacious in patients with unresectable HCC and unresectable hepatic metastatic disease, practical guidance on personalized dosimetry performance is still in progress.

Therefore, the most appropriate therapy for patients with HCC is liver transplantation, which addresses both the underlying cirrhosis and the HCC with a five-year survival rate of up to 85%, however, only 10% of patients are eligible [7].

Patients with HCC and liver cirrhosis face major challenges, as cancer must be under control, while patients remain on a waiting list for liver transplantation. During this waiting time, the progression of the tumor is unpredictable, resulting in an average dropout rate of ~25% [8] or as high as ~40% in 12 months [9,10]. Therefore, extended control of HCC is even more important for patients on the transplant waiting list to successfully reach liver transplants. Given the above, local therapy for HCC has been investigated as a bridge to liver transplant with the aim of decreasing tumor progression and reducing the risk of dropout rate from the waiting list.

Over the past decade, stereotactic body radiation therapy (SBRT) has emerged as a non-invasive, safe, and effective therapy for liver tumors providing local control and prolonging survival for many HCC patients who were not eligible for standard local regional treatment [11,12,13]. Image-guided SBRT for unresectable HCC sustained local control rates ranging from 75 to 100% in prospective phase I/II clinical trials [12,14]. Several prospective studies have shown SBRT as a highly effective ablative therapy for primary liver tumors and used as a bridge to liver transplant for inoperable patients with HCC [12,14,15]. Combined with appropriate diagnostic imaging, SBRT delivers the ablative radiation dose to liver tumor with conformal avoidance of residual functionally active hepatic parenchyma thus minimizing the risk of developing radiation-induced liver disease in patients with liver cirrhosis [13,16,17]. In this regard, biocompatible superparamagnetic iron oxide nanoparticles (SPION) have attracted a great deal of interest as contrast agents for magnetic resonance imaging (MRI) providing differential contrast enhancement imaging for functionally active, macrophage-infiltrated hepatic parenchyma and liver tumors for treatment planning of liver SBRT on MRI-Linac in patients with hepatic cirrhosis [18,19].

Ferumoxytol (Feraheme^®^) is a carbohydrate-coated ultrasmall SPION agent that is FDA-approved for the treatment of iron deficiency anemia. Because of its clearance through the reticuloendothelial system, ferumoxytol has been recently adopted for off-label clinical use as an MRI contrast agent for clinical imaging of liver parenchyma involving contrast uptake by hepatic macrophages (Kupffer cells) [20,21]. Once intravascularly injected, ferumoxytol nanoparticles stay trapped within hepatic Kupffer cells for several weeks causing T2-weighted signal loss within functionally active liver parenchyma [22]. Malignant tumors lacking Kupffer cells exhibit no signal change resulting in increased tumor-to-liver contrast difference allowing diagnostic quality MR imaging with contouring of liver tumors for precision targeting and functional hepatic parenchyma for guided avoidance during liver SBRT planning.

However, the radiomodulating properties of Feraheme on hepatic macrophages have never been directly investigated. Moreover, patients with primary and metastatic hepatic malignancies often present with iron-deficiency anemia due to chronic blood loss and require therapeutic Feraheme injections, raising the same question on the potential radiosensitizing effect of Feraheme on hepatic parenchyma when the liver is irradiated simultaneously with Feraheme injections. Although radiation-induced damage of Kupffer cells is well described [23], published results focusing on macrophage polarization after SPION treatment are highly inconsistent. For instance, previous studies reported that SPION polarizes macrophages into M1-phenotype [24,25], has no M1 polarization effect [26], or increases the production of anti-inflammatory IL-10 (i.p., M2 polarization) [27,28]. Furthermore, ferumoxytol upregulated macrophage polarization associated with pro-inflammatory Th1-type responses [29]. In vivo, ferumoxytol could inhibit tumor growth in mice, which was accompanied by tumor infiltration with pro-inflammatory M1 macrophages [29]. Finally, ferumoxytol might also cause tumor cell ferroptosis, an iron-dependent cell death, by triggering the transformation of infiltrating macrophages to the M1 phenotype [30]. However, the effect of ferumoxytol on macrophage polarization, cytokine release, and apoptosis under irradiation has never been investigated.

This study aimed to develop, optimize, and characterize the pre-clinical model allowing the determination of functional and phenotypic alterations of human macrophages treated with SPION at therapeutically relevant doses of ionizing radiation in vitro. Our results revealed that iron loading significantly improved monocyte and macrophage survival under low-dose irradiation. We also demonstrated that SPION supports the anti-inflammatory phenotype of human macrophages under radiation and that the effects of SPION depended on the duration of iron particle uptake and were radiation dose-dependent. Our data help to understand mechanisms of radiation-induced liver damage and provide the basis for the safe administration of Feraheme as an MRI contrast agent during SPION-enhanced MRI-guided liver SBRT to HCC in patients with hepatic cirrhosis awaiting a liver transplant.

## 2. Materials and Methods

### 2.1. Cell Cultures

THP-1 cells, a spontaneously immortalized monocyte-like cell line derived from the peripheral blood of a childhood case of acute monocytic leukemia (M5 subtype) [31] were purchased from ATCC (TIB-202). Cells were cultured in RPMI-1640 medium (Thermo Fisher Scientific, Waltham, MA, USA) supplemented with 0.2 mM of L-glutamine, 50 µ/mL of penicillin, 50 μg/mL of streptomycin, 10 mM HEPES (Invitrogen Life Technologies, Waltham, MA, USA), and 10% heat-inactivated fetal bovine serum (FBS, Gemini Bio-Products, West Sacramento, CA, USA) (complete RPMI-1640 medium) and maintained at 37 °C, 5% CO_2_ in a humidified tissue culture incubator. For macrophage differentiation, 1 × 10^6^ THP-1 cells/mL were cultured in a complete RPMI-1640 medium supplemented with 20 ng/mL recombinant human M-CSF (Peprotech Inc., Rocky Hill, NJ, USA) for five days. All cells were authenticated, mycoplasma tested, were contaminant-free, and used at low passage. For macrophage harvesting, trypsin was added to flasks until cells became detached and RPMI medium +10% FCS was added to neutralize the effects of the enzyme. The cell suspension was centrifuged at 400× *g* for 5 min and re-suspended in a lower volume of media. Cell number was determined using the hemocytometer and Trypan Blue staining (Sigma, Burlington, MA, USA).

### 2.2. Cell Proliferation Assay

To examine the cell proliferative activity of treated and control cells, 1 × 10^4^ cells per well were seeded in 96-well plates in a culture medium. Following appropriate treatments, 1 mg/mL MTT reagent (Sigma, Burlington, MA, USA) was added to each well for 4 h at 37 °C. The cells were then suspended in dimethyl sulfoxide for 3 h at 37 °C and detected using multimode microplate readers (BioRad, Hercules, CA, USA) at a wavelength of 540 nm. Quadruplicates were used in each experiment. The MTT (3-[4,5-dimethylthiazol-2-yl]-2,5 diphenyl tetrazolium bromide) assay is based on the conversion of MTT into formazan crystals by living cells, which determines mitochondrial activity and reflects cell proliferative activity.

### 2.3. Annexin V/PI Apoptosis Assay

Cells were collected, pelleted, and washed twice in PBS. The resulting pellets were resuspended in 100 μL of 1X Annexin V binding buffer and stained with 5 μL Annexin V-FITC and 10 μL propidium iodide (PI) (BioLegend, San Diego, CA, USA). Samples were kept at room temperature for 15 min and protected from light. After the incubation period, 400 μL of Annexin V binding buffer was added to each tube. Samples were analyzed immediately by flow cytometry (BD LSR II, Franklin Lakes, NJ, USA). Data were analyzed using FlowJo software V9 (FlowJo LLC., Ashland, OR, USA).

### 2.4. Flow Cytometry

After experimental and control treatments, cells were collected by gentle enzymatic detachment, resuspended, counted, and resuspended in flow cytometry staining buffer (PBS supplemented with 2% BSA) at 1 × 10^6^ cells/mL. Aliquots of cells were stained for 30 min at room temperature and protected from light with fluorescently conjugated antibodies (HLA-DR-FITC, CD86-PerCp-Cy5.5, CD11b-PE, CD14-APC from Biolegend) according to the concentrations indicated by the suppliers. Flow cytometric acquisition was performed on a Becton Dickinson LSR II instrument. Data were analyzed using FlowJo software V9 (FlowJo LLC.). A minimum of 10,000 events were acquired for each sample and the experiments were repeated three times independently.

### 2.5. Cytokine Secretion

Cell culture supernatants from M-CSF-differentiated macrophages were collected at the end of the polarization period and appropriate treatments. Secretion of cytokines was quantified using a multi-cytokine–chemokine panel 27plex assay, according to the manufacturer’s instructions (Bio-Rad, Hercules, CA, USA). The results were normalized to the total protein concentration in the cell culture supernatant samples. Data are derived from two independent experiments performed in triplicate.

### 2.6. Experimental Design

Monocytes or macrophages (200,000 cells/mL) were stabilized in cell cultures, harvested in fresh medium, and treated with ferumoxytol (510 mg/17 mL vial at a neutral pH, AMAG Pharmaceuticals, Inc., Waltham, MA, USA)—30 µg Fe/mL, 2 or 24 h, 37 °C. Ferumoxytol is a superparamagnetic iron oxide nanoparticle, 17–31 nm in diameter (topological polar surface area 74.6 Å^2^), coated with a low molecular weight semi-synthetic carbohydrate (polyglucose sorbitol and carboxymethyl ether shell) and having ~2000 magnetite iron (Fe_3_O_4_) molecules in the core with the relaxometric properties at 1.5 Tesla and 37 °C of r_1_ = 15 and r_2_ = 89 mM^−1^s^−1^ [20,32]. The coating material is about 1.7 nm thick (10 kDa in molecular weight) and provides ferumoxytol with a neutrally charged surface [30]. The concentration of ferumoxytol for our studies was selected based on previous in vitro data investigating the dose-dependent effects of SPION on monocytes and macrophages [24,33,34,35] and in vivo data on ferumoxytol pharmacodynamics and pharmacokinetics [12,36,37]. Uptake of iron oxide nanoparticles by macrophages has been well documented previously [22,32,35,38,39]. After iron uptake, cells were washed twice and treated with therapeutically relevant doses of gamma radiation (300, 500, 1000, or 3000 rad) (Gamma Cell 1000 Elite, Nordion International Inc., Ottawa, ON, Canada). Next, cells were cultured again in 96-well plates (MTT assay and cytokine expression) or 6-well plates (phenotyping and apoptosis assay) for 24 or 48 h and analyzed in different assays as indicated in individual result sections.

### 2.7. Statistical Analysis

For a single comparison of two groups, the Student *t*-test was used after the evaluation of normality. If data distribution was not normal, the Mann–Whitney rank-sum test was performed. For the comparison of multiple groups, ANOVA was applied. SigmaStat Software V4.0 was used for data analysis (SyStat Software, Inc., Chicago, IL, USA). For all statistical analyses, *p* < 0.05 was considered significant. All experiments were repeated at least two times. Data are presented as the mean ± standard error of the mean (SEM).

## 3. Results

### 3.1. Ferumoxytol Decreases Radiation-Induced Cell Death of Human Monocytes In Vitro

First, we tested how the uptake of iron by human monocytes affects their sensitivity to radiation-induced cell death. Figure 1 demonstrates that preincubation of cells with ferumoxytol for either two or 24 h before irradiation significantly increases their survival measured by Annexin V/PI staining for all tested doses of irradiation after 24 h. For instance, the level of cell death in monocytes incubated loaded with SPION for 24 h and irradiated by 500 rad decreased from 11.9 ± 2.1% to 7.7 ± 1.2% (Figure 1B, *p* < 0.05). We also revealed that 3000 rad killed more than 50% of cultured cells and this dose of radiation was omitted from further experiments.

To verify these results, all experiments were repeated with a prolonged time of cell culture after irradiation—cell death was assessed 48 h after cell irradiation (300–1000 rad). The results are demonstrated in Figure 2. Short (2 h, Figure 2A) loading of THP-1 cells with iron nanoparticles did not protect cells from higher doses of irradiation (500 and 1000 rad), while the protective effect was still detected at 300 rad: cells death significantly decreased from 5.5 ± 0.6% to 3.9 ± 0.3% (*p* < 0.05). Importantly, preincubation of cells with SPION for 24 h (Figure 2B) significantly decreased cell death induced by all tested doses of radiation. For instance, the level of cell death in monocytes incubated loaded with SPION for 24 h and irradiated by 500 rad decreased from 23.9 ± 4.2% to 14.8 ± 2.2% (*p* < 0.05).

Thus, these data suggest that human monocytes loaded with iron nanoparticles demonstrate increased resistance to radiation-induced cell death. This raises the next question about how iron uptake can alter the functional and phenotypic characteristics of human monocytes and macrophages.

### 3.2. Ferumoxytol Prevents Radiation-Induced Inhibition of Monocyte Proliferative Activity In Vitro

To determine how the uptake of iron particles alters monocyte proliferation after irradiation, cells were incubated with SPION for 2 and 24 h, irradiated (300, 500, and 1000 rad), and their proliferation was assessed 24 and 48 h later in an MTT assay. Figure 3 demonstrates that radiation dose-dependently inhibits the proliferative activity of human monocytes. Preincubation of cells with SPION for 2 h significantly abrogated this inhibitory effect of all tested doses of radiation seen after 24 h, although the protective effect persisted only for 300 rad if assessed 48 h after irradiation (Figure 3, upper panels). Importantly, significant iron-mediated protection was revealed for both early (24 h) and late (48 h) detection if monocytes were pre-treated with SPION for 24 h (Figure 3, lower panels). For instance, 0.55 ± 0.03 OD versus 0.61 ± 0.06 OD (*p* < 0.05) and 0.62 ± 0.07 OD versus 0.79 ± 0.05 OD (*p* < 0.05) for 500 rad detected in 24 and 48 h, respectively.

Thus, these data suggest that iron nanoparticles significantly inhibit the antiproliferative effect of radiation on human monocytes and that this effect was markedly stronger if cells were pre-treated with SPION for a prolonged time.

### 3.3. Modulation of Cytokine Production in Monocytes and Macrophages by SPION and Irradiation

Using the THP-1 monocytic cell line, it was reported that gamma radiation triggers monocyte differentiation toward the macrophage phenotype with increased expression of type I interferons and both pro- and anti-inflammatory macrophage phenotyping markers [40]. Therefore, we tested how SPION uptake can alter cytokine production by monocytes and macrophages upon irradiation. Undifferentiated and M-CSF-differentiated THP-1 cells were treated with SPION for 24 h (based on the results above), irradiated with 500 and 1000 rad, and cell-free supernatants were harvested 48 h later for determination of cytokine levels. Figure 4 shows cytokine production by control and treated monocytes. These results revealed that ferumoxytol uptake by human monocytes downregulates expression of pro-inflammatory chemokines MIP-1α (macrophage inflammatory protein 1α), MIP-1β (CCL4), and RANTES (CCL5) but does not significantly alter cytokine expression under radiation conditions. Only radiation (500 rad) induced upregulation of monocyte chemoattractant protein-1 (MCP-1/CCL2), one of the key chemokines that regulate migration and infiltration of monocytes/macrophages, was abrogated by SPION, but this finding should be further investigated. Of note, IL-2, IL-4, IL-5, IL-6, IL-7, IFN-γ, IL-10, IL-12 (p70), IL-13, IL-1β, G-CSF, FGF-2, GM-CSF and PDGF levels were less than 1 ng/mL.

Next, using a similar experimental design, we tested cytokine expression in macrophages differentiated from THP-1 cells and treated with SPION and irradiation. Results in Figure 5 show that macrophages produce significantly higher levels of cytokines, when compared with monocytes, and are much more sensitive to iron uptake and radiation treatment. For instance, ferumoxytol uptake by macrophages significantly upregulated the expression of MCP-1 (CCL2) in control and irradiated cells (*p* < 0.05). SPION also reversed the effect of radiation (500 rad) on the expression of IL-1RA, IL-8, VEGF, CCL5, and TNF-α. Interestingly, if irradiation upregulated cytokine expression (IL-1RA, TNF-α, CCL5), iron particles downregulated their expression. However, if irradiation decreased cytokine release from macrophages (IL-8, VEGF), iron increased cytokine production. Furthermore, strong stimulation of MCP-1, MIP-1α, and MIP-1β expression in macrophages by ferumoxytol upon cell irradiation should be further investigated. Also, expression of IL-2, IL-4, IL-5, IL-6, IL-7, IFN-γ, IL-10, IL-12 (p70), IL-13, IL-1β, G-CSF, FGF-2, GM-CSF and PDGF was not observed.

Together, these results suggest that iron uptake by human monocytes and macrophages may alter their sensitivity to irradiation-induced cytokine expression.

### 3.4. Ferumoxytol Changes the Phenotype of Monocytes/Macrophages Altered by Radiation

Finally, we asked whether iron nanoparticles could alter monocyte and macrophage phenotype in response to gamma irradiation. First, we showed that 1000 rad treatment of THP-1 cells decreased the percentage of CD11b^+^ CD14^+^ cells from 92.3 ± 5.0% to 65.2 ± 6.2% (*p* < 0.05) 48 h after irradiation but the preincubation with SPION for 24 h before irradiation kept the percentage of double-positive cells at 82.6 ± 7.3% level. Furthermore, while 1000 rad increased CD11b^neg^ CD14^neg^ cells from 2.3 ± 0.1% to 21.7 ± 4.6% (*p* < 0.05), SPION reversed this effect to 7.1 ± 1.1% (*p* < 0.05). Importantly, a similar anti-radiation effect of SPION was also seen in macrophage cultures: while 1000 rad significantly upregulated CD11b^+^CD14^+^ cells from 56.7 ± 7.4% to 75.5 ± 8.2%, iron nanoparticles prevented this increase to 62.3 ± 6.3% level (*p* < 0.05). As an example, Figure 6 demonstrates that expression of HLA-DR in human macrophages decreased under irradiation and, interestingly, uptake of iron particles before cell irradiation decreased even more: 94.1 ± 7.2% versus 75.4 ± 9.1% (*p* < 0.05). Furthermore, radiation upregulated the expression of CD86 in macrophages, and SPION augmented it further: 3.9 ± 0.2% versus 18.5 ± 2.2% (*p* < 0.05) (Figure 6). As a control, Figure 6 shows that SPION does not change the expression of HLA-DR and CD86 on preloaded macrophages in control non-irradiated cultures.

Thus, these data suggest that SPION can attenuate polarization of macrophages induced by radiation, although the effect depends on the dose of radiation and phenotypic markers.

## 4. Discussion

MRI-guided radiotherapy on a hybrid MR-Linac (a magnetic resonance-guided linear accelerator) is a rapidly evolving new technology allowing superior soft tissue imaging for liver SBRT compared with conventional CT-guided radiotherapy. Reliable identification of liver tumors and functional hepatic parenchyma on MR-Linac has a direct impact on the quality of radiotherapy planning and treatment outcomes. However, using repeated MRI with conventional IV contrast, such as gadolinium chloride for daily image-guided radiotherapy carries the potential risk of toxicity and is contraindicated in patients with impaired kidney function.

Ferumoxytol (Feraheme^®^), as a SPION-based contrast agent with unique pharmacological, metabolic, and imaging properties, may play a crucial role in the future MR liver imaging [30,39] with important safety features: (a) it can be safely administered in the population of patients with impaired renal function in whom gadolinium-based MRI contrast agents are contraindicated; (b) the use of ferumoxytol is not associated with concerns of a long-term accumulation from repeated applications, such as is the case with brain deposition of gadolinium-containing agents [40,41].

Ferumoxytol is eventually taken up by tissue-resident macrophages/the reticuloendothelial system in the liver, spleen, bone marrow, and lymph nodes, and this uptake mechanism is being extensively explored for the enhanced MR imaging approach for tumors, vascular lesions, and lymph nodes [20]. For instance, the utilization of ferumoxytol as an off-label contrast agent has been recently reported to increase the detection rate of colorectal cancer liver metastases and may aid in preoperative decision-making [41].

Lately, superparamagnetic iron oxide nanoparticles have been used to diagnose focal liver lesions and the progression of fibrosis in steatohepatitis by the analysis of iron nanoparticles in liver Kupffer cells. It was reported that alteration of Kupffer cell phagocytic function evaluated with SPION-MRI correlated with the severity of non-alcoholic steatohepatitis [42]. Furthermore, analysis of the diagnostic value of SPION/MR imaging for the characterization of focal liver lesions, both primary and metastatic, in patients with cancer and hepatic cirrhosis revealed a diagnostic incremental value of using iron oxide particles [43]. The usefulness of SPION was based on the high uptake of the SPION by the Kupffer cells: a shortage of T2W signal was seen in the volumes of functional hepatic parenchyma diffusely infiltrated by Kupffer cells, with no signal changes in hepatic lesions lacking Kupffer cells [44].

As the use of ferumoxytol, a novel ultrasmall SPION formulation, as a contrast agent for MRI is constantly increasing, it is critical to understand its radiomodulating effects on liver parenchyma infiltrated by hepatic resident macrophages in a setting of SPION-enhanced MRI-guided SBRT to primary and metastatic liver tumors.

Iron oxide nanoparticles have been reported to augment a “pro-inflammatory” immune cell phenotype in macrophages and their antitumor potential [29,45,46]. For instance, the growth of breast adenocarcinomas in mice was markedly repressed by ferumoxytol, which was associated with the alteration of pro-inflammatory M1 macrophages in the tumor tissues [29]. In addition to stimulating macrophages, ferumoxytol can reduce the immunosuppressive function of myeloid-derived suppressor cells (MDSC) known to play a key role in the formation of immunosuppressive tumor microenvironment and resistance to anti-cancer therapy [47].

The radiosensitizing potential of SPION is intensively investigated revealing the diversity of results among the multiple research groups [48]. Even though radiation-induced modulation of monocytes and macrophages has been described [40,49,50] to the best of our knowledge, there are no data describing the effect of ferumoxytol on human monocytes and macrophages under clinically relevant doses of radiation.

Here we demonstrated that preloading of human monocyte cell line with ferumoxytol significantly decreased radiation-induced cell death of human monocytes in vitro. The effect was revealed using doses of gamma radiation within the liver SBRT clinical dose range. Although Wu et al. demonstrated that Fe_3_O_4_ nanoparticles could reduce macrophage viability via activation of ferroptosis after 48 h through the upregulation of p53 [51] we did not observe the effect of ferumoxytol on THP-1 cell death in vitro. However, as expected, we observed radiation-induced cell death, which was significantly attenuated by cell preincubation with iron oxide nanoparticles. Our data were further confirmed by the demonstration that ferumoxytol prevented radiation-induced inhibition of monocyte proliferative activity in vitro. The inhibitory effect of gamma radiation on primary macrophages has been described [52], but our data demonstrated for the first time that monocytes preloaded with ferumoxytol displayed increased resistance to the anti-proliferative effect of radiation on human monocytes. Interestingly, Teresa Pinto et al. reported that irradiated (200 rad/fraction/day for a week) human monocyte-derived macrophages remained viable and metabolically active, and increased Bcl-xL expression evidenced the promotion of pro-survival activity [49]. It would be interesting to assess the potential role of SPION in macrophage longevity in this experimental model.

Furthermore, we investigated the cytokine production by human monocytes and macrophages treated with ferumoxytol and radiation. The pattern of altered cytokine expression did not allow for a conclusion about a definite pro- or anti-inflammatory phenotypic polarization of treated cells, which was not a surprising finding based on previously published contradicting data. Our observation of the absence of strong pro-inflammatory polarization of monocytes loaded with iron nanoparticles agrees with Raynal et al. who reported that uptake of even high concentrations of SPION (ferumoxide or ferumoxtran-10) by activated THP-1 cells caused a very low IL-1 expression [53]. However, Laskar et al. reported that SPION induced a phenotypic shift in THP-1-derived M2 macrophages towards a high CD86+ and high TNF-α+ macrophage subtype [25]. Nonetheless, irradiation of human monocyte-derived macrophages with 200, 600, or 1000 rad has been reported to result in reduced expression of anti-inflammatory genes [49]. Because of the visual increase in pro-inflammatory macrophage markers CD80, CD86, and HLA-DR, but not TNF-α and IL-1β after 1000 rad cumulative doses, with downregulated anti-inflammatory markers and IL-10 expression, the authors concluded about the modulation towards a more pro-inflammatory phenotype. Interestingly, we observed that preloading of THP-1-derived macrophages with ferumoxytol downregulated the expression of HLA-DR and upregulated the expression of CD86. The most interesting observation is that radiation may augment both of these pathways.

Here It is important to understand that multiple controversies between published data can be explained by other results demonstrating that different types of SPION particles display differential effects on macrophages due to their size, polarity, cover layers, and cytotoxic properties [48,54,55]. Next, the uptake and effect of SPION particles also depend on the subset of monocytes and macrophages used for evaluation, including their state of activation and polarization, culture conditions, source, and species [25,27,28,56]. Similarly, the alteration of monocytes and macrophages under irradiation conditions also depends on the type of radiation, accumulative dose, radiation schedule, and type of cells used for the assay [49,52,57,58]. Additional evaluation of ferumoxytol and radiation combination on primary human monocytes and macrophages harvested from healthy donors and patients with different diseases is needed to confirm our initial experience.

In conclusion, we have demonstrated that ferumoxytol, in addition to being an FDA-approved iron oxide nanoparticle agent for the treatment of iron-deficiency anemia, possesses unique radiomodulating effects on human monocytes and macrophages irradiated within therapeutically relevant doses of gamma radiation. Ferumoxytol affected both human monocytes and macrophages, increased the resistance of monocytes to radiation-induced cell death, alleviated inhibition of cell activity, and supported the anti-inflammatory phenotype of human macrophages under clinically relevant doses of radiation. Its effect on human monocytes depended on the duration of iron particle uptake and was radiation dose-dependent. In future studies, we expect to investigate the radiomodulating effects of ferumoxytol on primary human monocytes and Kupffer cells, focusing on the analysis of how SPION-preloaded macrophages regulate the viability and function of primary human hepatocytes within the normal liver, in the presence of tumors, and under cirrhotic microenvironments in vitro and in vivo. These studies will further investigate the diagnostic and therapeutic properties of ferumoxytol and provide new insight into the limitations and emerging applications of SPIONs in biomedicine.

## Figures and Tables

**Figure 1 cancers-16-01318-f001:**
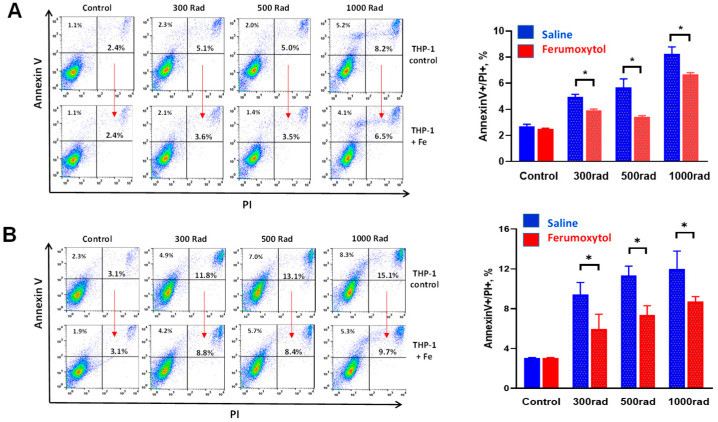
Ferumoxytol decreases radiation-induced apoptosis of human monocytes in vitro. THP-1 monocytes were treated with SPION (30 µg Fe/mL) for two (**A**) and 24 (**B**) hours, washed, irradiated by 300–1000 rad, and analyzed in Annexin V/PI assay 24 h later as described in M&M. Both Annexin V+/PI− (early apoptosis) and Annexin V+/PI+ (all dead) cells were analyzed. Treatment with saline served as a control. All samples were tested in triplicates in each experiment. The left panels represent the results of representative experiments, while the right panels summarize the results of 3–4 independent experiments. Error bars indicate ±SEM of 3–4 independent replicates. *, *p* < 0.05 (Student *t*-test, *n* = 3–4).

**Figure 2 cancers-16-01318-f002:**
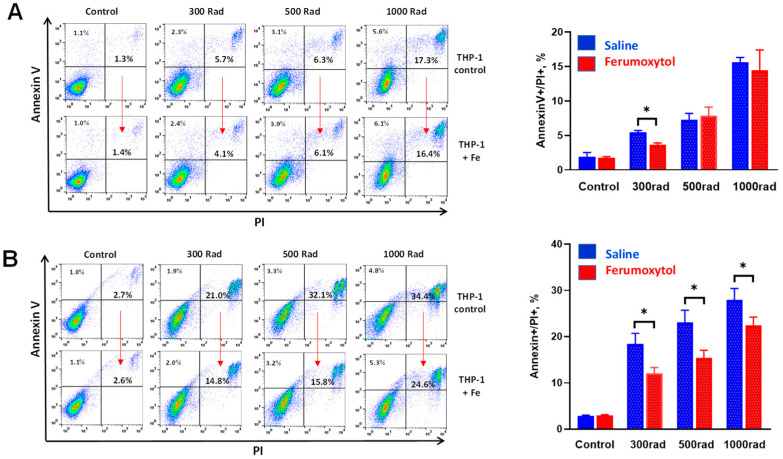
Ferumoxytol decreases radiation-induced apoptosis of human monocytes in vitro. THP-1 monocytes were treated with SPION (30 µg Fe/mL) for 2 (**A**) and 24 (**B**) hours, washed, irradiated by 300–1000 rad, and analyzed in Annexin V/PI assay 48 h later as described in M&M. Both Annexin V+/PI− (early apoptosis) and Annexin V+/PI+ (all dead) cells were analyzed. Treatment with saline served as a control. All samples were tested in triplicates in each experiment. The left panels represent the results of representative experiments, while the right panels summarize the results of 3–4 independent experiments. Error bars indicate ±SEM of 3–4 independent replicates. *, *p* < 0.05 (Student *t*-test, *n* = 3–4).

**Figure 3 cancers-16-01318-f003:**
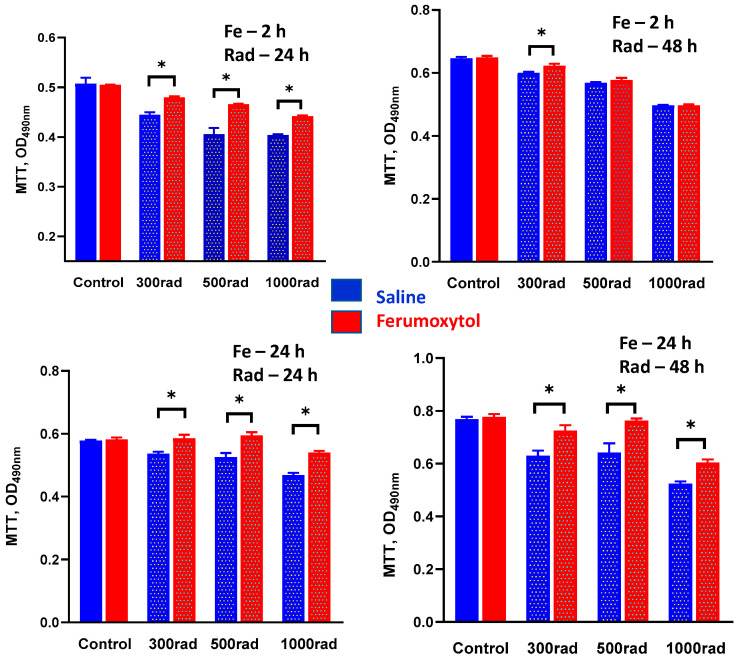
Ferumoxytol prevents radiation-induced inhibition of human monocyte proliferative activity in vitro. THP-1 monocytes were treated with SPION (30 µg Fe/mL) for 2 (**upper panels**) and 24 (**lower panels**) hours, washed, irradiated by 300–1000 rad, and analyzed in MTT assay 24 h (**left panels**) and 48 h (**right panels**) later as described in M&M. Treatment with saline served as a control. All samples were tested in triplicates in each experiment. Error bars indicate ±SEM of four independent replicates. *, *p* < 0.05 (Student *t*-test, *n* = 4).

**Figure 4 cancers-16-01318-f004:**
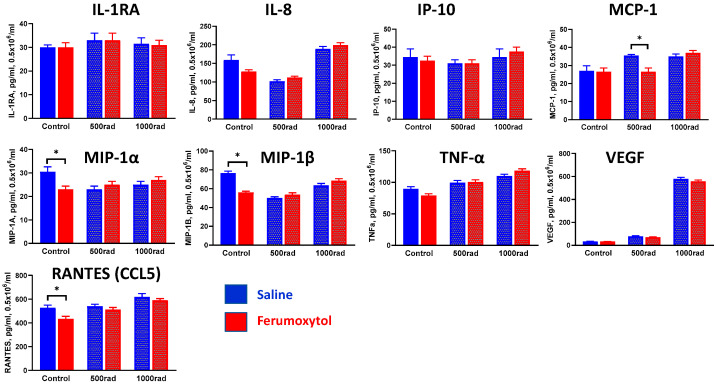
Ferumoxytol and radiation modulate cytokine expression in human monocytes in vitro. THP-1 monocytes were treated with SPION (30 µg Fe/mL) for 24 h, washed, irradiated by 500 or 1000 rad, and cultured for an additional 48 h before supernatants were collected for cytokine assessment as described in M&M. Treatment with saline served as a control. All samples were tested in triplicates in each experiment. Results are shown as mean ± SEM. *, *p* < 0.05 (Student *t*-test, *n* = 3).

**Figure 5 cancers-16-01318-f005:**
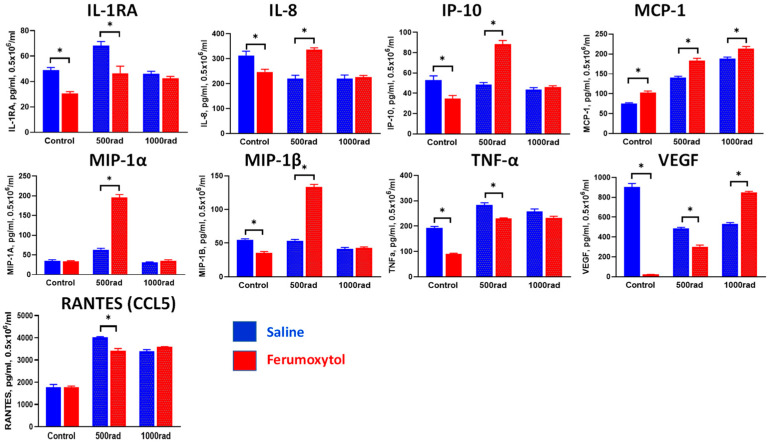
Ferumoxytol and radiation modulate cytokine expression in human macrophages in vitro. THP-1 monocytes were treated with M-CSF (20 ng/mL) for five days, then loaded with SPION (30 µg Fe/mL) for 24 h, washed, irradiated by 500 or 1000 rad and cultured for an additional 48 h with M-CSF (10 ng/mL) before supernatants were collected for cytokine assessment as described in M&M. Treatment with saline instead of SPION served as a control. All samples were tested in triplicates in each experiment. Results are shown as mean ± SEM. *, *p* < 0.05 (Student *t*-test, *n* = 3).

**Figure 6 cancers-16-01318-f006:**
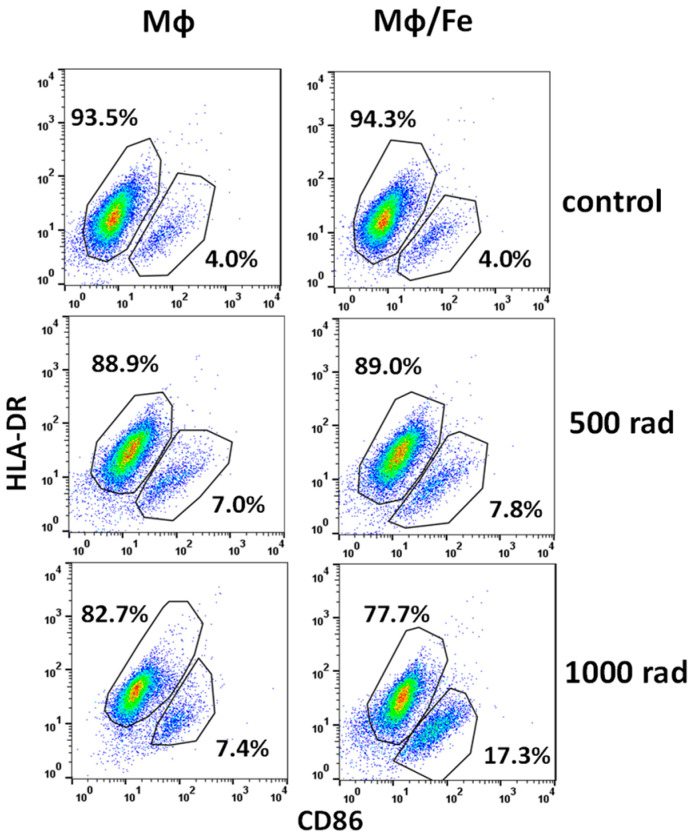
Ferumoxytol changed phenotypic alterations of monocytes and macrophages induced by radiation in vitro. Undifferentiated and macrophage-differentiated (M-CSF, 20 ng/mL, five days) THP-1 cells were treated with medium (control) or ferumoxytol (30 µg Fe/mL, 24 h) and irradiated by 500 and 1000 rad. Cells were then cultured for an additional 48 h with medium (monocytes) or M-CSF, 10 ng/mL (macrophages) before their phenotype was assessed by flow cytometry as described in M&M. Results from a representative experiment are shown. Mф, macrophages.

## Data Availability

The original contributions presented in the study are included in the article, further inquiries can be directed to the corresponding author(s).

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
