# Peer review of "Radiomodulating Properties of Superparamagnetic Iron Oxide Nanoparticle (SPION) Agent Ferumoxytol on Human Monocytes: Implications for MRI-Guided Liver Radiotherapy"

_cancers, 2024, doi:10.3390/cancers16071318_

Round 1

Reviewer 1 Report

Comments and Suggestions for Authors

The authors used FDA-approved iron oxide nanoparticles to study the radiomodulating properties of human monocytes and macrophages with therapeutical doses in cell levels. Results showed ferumoxytol can increase the resistance of monocytes to radiation-induced cell death and inhibition of cell activity. The results are dose dependent. The manuscript is well written, and the study provided preliminary evidence in SPINO-enhanced MRI-guide liver radiation therapy. The authors need to address the following issues before the manuscript is accepted for publication in Cancers:

  1. In Figure 1 and Figure 2, the Annexin V/PI assay was performed 24 and 48 h, respectively. Please explain why different time points were selected.
  2. All the cellular studies use the same concentration of SPINO (30 µg Fe/ml). In addition to different radiation doses, do the authors expect the iron loading dose to show any effect on the results?
  3. Please check the reference format in the manuscript. Some refs. are cited after the stop.
Comments on the Quality of English Language

The manuscript is well written.

Reviewer 2 Report

Comments and Suggestions for Authors

This study "Radiomodulating properties of superparamagnetic iron oxide nanoparticle (SPION) agent ferumoxytol on human monocytes: implications for MRI-guided liver radiotherapy" illustrates the role of nanoparticles targeting iron receptors on Kupffer cells in liver tumour cells; nanoparticles have recently been increasingly used in various fields of medical diagnostics and therapy.

- The introduction part of the paper is informative and provides the background to the aims of the study. 

- The methods part contains a detailed description from the cell culture to the assays used to analyse cell proliferation and apoptosis. 

- All results are presented and discussed in detail in the "Discussion" section. The authors outlined the similarities and differences between their results and the results of other studies in this field.

Nevertheless, I have a few comments: 

- Please briefly discuss selective internal radiation therapy (SIRT), as it is also an important focal therapy for non-resectable liver tumours and metastases.

- Some abbreviations are not previously defined, such as SEM 

Comments on the Quality of English Language

- The entire paper needs to be checked for English by a native English speaker.

Reviewer 3 Report

Comments and Suggestions for Authors

The authors presented the paper "Radiomodulating properties of superparamagnetic iron oxide nanoparticle (SPION) agent ferumoxytol on human monocytes: implications for MRI-guided liver radiotherapy "

1) A reference list for Introduction section should be improved. Some 2022-2024 years references should be inserted and discussed to show the progress in the area.

2) Section 2. Please, insert the most important materials such as magnetite nanoparticles resource. Moreover, the data of radiation may be better divided into one separate subsection. Part of the section (lines 166-178) Experimental design is more relevant for the Introduction or Results and Discussion section.

3) I see that you have studied your experiment at only one concentration of nanoparticles. Have you studied your effect using various dose of SPIONs?

4) Figure 1 caption. Both Annexin V+/PI- (early apoptosis) and Annexin V+/PI+ (all dead) cells were analyzed. However on the Figure I see only V+/PI+ on the right.

5) It will be excellent to present the Conclusion section highlighting paper novelty and future perspectives.

Minor 

smth with text style, lines 27, 53-55, 72, 76-79, 164-165, 180

Comments on the Quality of English Language

 Minor editing of English language required
